# Dietary Micronutrient Intake During Pregnancy Is Suboptimal in a Group of Healthy Scottish Women, Irrespective of Maternal Body Mass Index

**DOI:** 10.3390/nu17030550

**Published:** 2025-01-31

**Authors:** Eleanor M. Jarvie, Julie A. Lovegrove, Michelle Weech, Dilys J. Freeman, Barbara J. Meyer

**Affiliations:** 1School of Medicine, Dentistry and Nursing, University of Glasgow, Scotland Q12 8QQ, UK; elliejarvie1@gmail.com; 2Department of Food and Nutritional Sciences, University of Reading, Reading RG6 6DZ, UK; j.a.lovegrove@reading.ac.uk (J.A.L.); m.weech@reading.ac.uk (M.W.); 3School of Cardiovascular and Metabolic Health, University of Glasgow, Scotland Q12 8QQ, UK; dilys.freeman@glasgow.ac.uk; 4School of Medical, Indigenous and Health Sciences, University of Wollongong, Wollongong, NSW 2522, Australia

**Keywords:** pregnancy, macronutrient, micronutrient, harmonised average requirements, BMI

## Abstract

**Background/Objectives**: A balanced nutritious diet is vital during pregnancy for both the mother and the baby. The aims of this longitudinal study were to (1) determine any differences in macro- and micronutrient intakes in a group of UK women during pregnancy (and in the post-partum period) who were overweight or obese (BMI mean (SD) 31.1 (2.9)) at antenatal booking appointment compared with women who were within the ideal BMI range (BMI mean (SD) 22.1 (1.9)) and (2) determine the proportion of women who met the Harmonized Average Requirements (H-AR) during pregnancy. **Methods**: Forty-two participants attended four clinic visits: three during pregnancy, one in each trimester (V1, V2, and V3), and one 12 weeks post-partum (V4). Dietary intake was assessed by 24 h diet recall and analysed using DietPlan6. **Results**: There were no differences in energy and macronutrient intakes between overweight/obese and lean women. During pregnancy, the overweight/obese women consumed a mean (SD) of 3238 (941) sodium (mg per day), which was approximately 10% higher compared to 2934 (732) sodium (mg per day) in the lean group (*p* = 0.015). Dietary and supplemental intakes of the sodium to potassium ratio was 21% higher in overweight/obese women compared to the lean women, *p* = 0.0031 (mean (SD) of 1.17 (0.35) versus 0.93 (0.28), respectively). Virtually all women did not meet the H-AR for niacin, folate, and vitamin D through dietary intake alone. **Conclusions**: The ‘eat better and not more’ message during pregnancy is supported.

## 1. Introduction

Nutrition is important during pregnancy as the mother needs to meet the demands of the growing foetus whilst maintaining optimal maternal nutritional status. There are general dietary guidelines, including the UK Eat Well Guide [1], the Australian Dietary Guidelines [2], and the US Dietary Guidelines [3], as well as recommendations from the International Federation of Gynaecology and Obstetrics (FIGO) and Royal College of Obstetricians and Gynaecologists (RCOG) [4], that support the consumption of a healthy, balanced diet throughout pregnancy and in the post-partum period. These recommendations provide the necessary energy and macro- and micronutrient intakes for both the mother and foetus. The United Kingdom (UK) Dietary Reference Values (DRVs) clearly outline recommended intakes for macro- and micronutrients for the general population as well as for pregnant women [5].

There are certain nutrients of relevance to pregnancy that must be obtained from the diet, and a notable example is folate, which is an essential nutrient and must be obtained through the diet as humans cannot synthesise folate. It is recognised that inadequate dietary folate intakes can lead to foetal neural tube defects (NTDs), such as Spina Bifida, and that supplementation of 4.mg/d of folate in the first trimester of pregnancy results in a 70% reduction in the incidence of NTDs, as shown by the landmark definitive MRC Vitamin Study [6]. Since this research, the actual dose of folate recommendation to prevent NTDs was lowered and is now 400 µg/d in the UK [5], 600 µg/d in Australia [7], and 520 µg/d, as proposed by the Harmonized Nutrient Reference Values for populations [8].

The quality of food intake is important to provide the necessary micronutrients for optimal health. Recommendations from assembled experts on the health benefits of consuming nutritious food before, during, and after pregnancy suggest eating better but not more [9]. A multi-national scoping review identified that vegetable intake during pregnancy falls below recommendations worldwide for women of all body weights and needs to be addressed [10]. It was estimated that pregnant women consume approximately 27% of their energy from ultra-processed foods (UPFs) determined by the NOVA classification. This is similar to the general population, although women with obesity pre-pregnancy with the lowest blood haemoglobin levels were reported to have significantly higher intakes from UPFs compared to normal weight women [11]. This difference in UPF consumption may contribute to differences in macro- and micronutrient intakes in pregnant women of varying degrees of adiposity. General dietary advice is provided during the course of antenatal care [12] but is often not specific to the overweight/obese population. Guidance on the management of overweight/obese pregnancies [13] highlighted the need for advising women about the importance of a healthy diet and exercise during pregnancy in order to avoid excessive weight gain and gestational diabetes mellitus. The only specific nutritional guidance for overweight/obese woman regarded the increased supplementation of folic acid and vitamin D (5mg and 10 µg daily, respectively) and did not discuss other aspects including fat and complex carbohydrate consumption.

Maternal dietary intake can have a marked effect on the growing foetus and the newborn. A review by Hovedenak and Haram [14] described the impact of vitamin and mineral deficiencies on pregnancy outcomes. Examples include maternal iron deficiency that resulted in reduced neonatal iron stores and birth weight, and vitamin D supplementation in women with vitamin D deficiency in the third trimester, which was shown to be safe up to 4000 IU [15], improving vitamin D and calcium status, and thereby protecting skeletal health [16]. In the Healthy Start Study [17], 1410 pregnant girls and women (aged 16 years and older, and less than 24 weeks of gestation) were recruited from the University of Colorado Hospital (USA). The study reported that higher maternal intake of total fat, saturated fat, unsaturated fat, and total carbohydrate, but not protein, was associated with increased neonatal adiposity [18]. A further study showed associations of higher maternal intake of total fat, saturated fat, and excessive added sugar with increased infant percent body fat at 6 months [19]. There was a recent systematic review and meta-analysis of macronutrient and energy intake during pregnancy, which included 135,566 pregnant women from America, Eastern Mediterranean, Western Pacific region, Europe, and Southeast Asia and it showed that the mean intakes from energy, carbohydrate, fat, and protein were 2036kcal/day (8519kJ/day), 262g/day, 74g/day, and 78g/day, respectively [20]. However, there is limited evidence on micronutrient intake in pregnant women, with one study reporting low intakes in overweight/obese pregnant women [21], although the study lacked a comparator healthy weight control group. Hence, there is a scarcity of research that reports the dietary intakes of micronutrients in pregnant women of varying levels of pre-pregnancy weight status.

The aims of this longitudinal study were to (1) determine any differences in macro- and micronutrient intakes in healthy (without any metabolic disease) UK women during pregnancy (and in the post-partum period) who were overweight or obese at antenatal booking appointment compared with women who were within the ideal BMI range and (2) determine the proportion of women who met Harmonized Average Requirements (H-AR) during pregnancy.

## 2. Materials and Methods

### 2.1. Study Participant Recruitment and Ethics Approval

The Lipotoxicity in Pregnancy Study [22] was conducted in National Health Service Greater Glasgow and Clyde maternity units between March 2010 and November 2011. The study was approved by the West of Scotland Research Ethics Committee (reference 09/S0701/105) and participants provided informed consent. Women were recruited at their first antenatal appointments and represented healthy Caucasian women between the ages of 16–40 years with no significant past medical history. Women were excluded from participating in the study if they had existing metabolic disease such as diabetes mellitus, thyroid disease, polycystic ovarian syndrome, or cardiovascular disease. Multiple pregnancies and pregnancies conceived with assisted conception were excluded as these were considered higher risk pregnancies for vascular and metabolic complications. Furthermore, women who developed obstetric antenatal complications such as hypertension or gestational diabetes mellitus throughout the study were retrospectively excluded. The BMI ranges for the lean group was from 17.8 to 25.4 kg/m^2^ and the overweight/obese group was from 27.1 to 36.6 kg/m^2^.

### 2.2. Dietary Intake Data Collection and Nutrient Analysis

Forty-two participants attended four clinic visits: three during pregnancy, one in each trimester (V1, V2, and V3), and one 12 weeks post-partum (V4). The mean (SD) for sampling gestations is shown in Table 1. At each study visit, either the researcher (E.J.) or a research nurse administered a 24 h dietary recall using the multiple pass method [23]. The multiple pass method is a five-step process for collecting dietary data that involves multiple passes through a 24 h period: (1) quick list: the subject lists all foods and beverages consumed without interruption; (2) forgotten foods: the subject is asked about foods they may have forgotten to list; (3) time and occasion: the subject records the time and occasion for each food; (4) detail cycle: the subject provides detailed information about each food, including the amount consumed, cooking methods, and brands; and (5) final probe: the subject is asked if they consumed anything else. The recall recorded everything that the participant ate and drank in the 24 h preceding the study visit, including dietary supplements, and was administered according to protocols used for the Food Standard Authority’s Low Income Diet and Nutrition Survey. The multiple pass method is not time consuming (approximately 15–20 min to complete) and is widely implemented in different demographic study groups. Dietary intake data were entered into the Dietplan 6 analysis package (Forestfield Software Ltd., West Sussex, UK) to obtain total energy and macronutrient and micronutrient intakes, which were exported from this database into Microsoft excel spreadsheets. Given the limitations of the multiple pass 24 h dietary recall, an average of V1, V2, and V3 dietary intakes were calculated to obtain a more representative dietary intake during pregnancy. The macronutrient data were expressed as percent of total energy. The micronutrient data were reported as actual intakes (with and without supplemental intakes). Dietary fibre intake and the micronutrient intakes were corrected for energy intake by dividing the average intake, V1, V2, V3, by the average energy intake, V1, V2, V3, prior to statistical analysis. Similarly, the V4 intake was divided by the V4 energy intake prior to statistical analysis.

### 2.3. Meeting Harmonised Average Requirement (H-AR) Criteria

The UK DRVs are estimates of the daily amounts of nutrients that healthy people need. They are used to assess the safety and adequacy of nutrient intake in a population [5]. H-AR is a nutrient intake level that meets the needs of half of a population. It is used to estimate how many people in a group are deficient in a nutrient [8]. The H-AR intakes for pregnant women were used to determine if women met or exceeded the H-AR for each micronutrient (with the exception of sodium, where intakes should be lower than the recommended intakes) and the data were recorded as yes or no, i.e., nominal data.

### 2.4. Statistical Analysis

For the Lipotoxicity in Pregnancy Study [22], the numbers recruited were based on power calculations for differences in the anthropometric measures [24], energy expenditure [25], endothelial function [26], and lipotoxic measures [27] but not for biological assays. The number of participants required was based on the mean obese measurement of the described parameter minus the mean lean measurement divided by the mean lean. This is described at the standardised delta. The second calculation required for the power was the sigma of the lean group divided by the mean lean. These results were then exported to Minitab vs16 where numbers needed to recruit were based on a 2 sample t power calculation. As the largest number needed to recruit was based on endothelial function (*n* = 24), it was decided to aim to recruit 30 women to each BMI group in order to detect a difference and cover any study drop-outs.

Values are expressed as the mean (SD) unless otherwise specified. A comparison of dietary data during and after pregnancy on the lean and overweight/obese women was made using a 2 × 2 analysis for during pregnancy versus post-partum in lean and overweight/obese in a mixed model with repeated measures. Chi square with Pearson correlation were used to analyse nominal variables. Statistical analyses were conducted using JMP 17.0.0 (622753) Copyright © 2022 and significance was set at *p* < 0.05.

## 3. Results

### 3.1. Study Participants’ Characteristics (Table 1)

A total of 30 lean and 22 overweight/obese women were recruited at the first antenatal clinic appointment to take part in the study and 26 lean and 16 overweight/obese women completed the study. All women (*n* = 42) were Caucasian and non-smokers, and their demographic data according to obesity group are shown in Table 1. On average, the overweight/obese women (*n* = 16) were 26kg heavier (*p* < 0.0001) and their booking BMI (V1) was 29% higher (*p* < 0.0001) than the comparator lean group (*n* = 26), as would be expected for the selection criteria. The lean group increased their body weight by approximately 7–8% across the trimesters, whereas the overweight/obese group increased their body weight by approximately 5% across the trimesters (Table 1). Both groups had blood pressure within the normal range, but the overweight/obese group had significantly higher systolic (*p* = 0.014) and diastolic (*p* = 0.016) blood pressure compared to the lean group. There were no differences between the groups in age, education level attained, socioeconomic status, alcohol consumption, height, and sampling time points (Table 1).

### 3.2. Comparison of Energy and Macronutrient Intakes in Pregnant Women with Published Meta-Analysis [20] (Appendix A Table A1)

The macronutrient intakes expressed as grams per day are shown in Appendix A Table 1 and there were no differences between the overweight/obese and lean groups (Appendix A Table A1). In comparison to a meta-analysis of energy and macronutrient intake [20], the current study energy intake was 6% (lean group) and 9% (overweight/obese group) higher than the mean of the meta-analysis of 54 studies [20]; primarily due to increased fat (13% lean group and 19% overweight/obese group) and carbohydrate intake (8% lean group and 10% overweight/obese group) with protein intakes being similar to those published in the meta-analysis [20].

### 3.3. Comparison of Macronutrient Intakes Between Lean and Overweight/Obese Pregnant Women During Pregnancy and in the Post-Partum Period Is Shown in Table 2

Regardless of BMI group, there was a trend for lower energy intakes in the post-partum period relative to during pregnancy (*p* = 0.051).

There were no significant differences in macronutrient intakes (expressed as percent of total energy) in terms of group effects (i.e., between lean and overweight/obese groups) or time effects (i.e., during pregnancy versus post-partum period), as shown in Table 2. There were two significant interaction terms across time and between the two groups: notably in carbohydrate and starch intakes. The dietary carbohydrate intake was 10% lower in the post-partum period compared to during pregnancy in the overweight/obese group whereas it was 2% higher in the lean group (*p* = 0.037). The dietary starch intake was 11% lower in the post-partum period compared to during pregnancy in the overweight/obese group whereas it was 4% higher in the lean group (*p* = 0.044).

**Table 2 nutrients-17-00550-t002:** Macronutrient intakes during pregnancy (average of V1, V2, and V3 intakes) and in post-partum period in lean versus overweight/obese women ^#^.

Macronutrient	Average Intake During Pregnancy Lean (*n* = 26)	Average Intake During Pregnancy Overweight/Obese (*n* = 16)	Post-Partum Intake Lean (*n* = 25)	Post-Partum Intake Overweight/Obese (*n* = 15)	*p* Lean vs. Obese	*p* Time	*p* Time × Lean/Obese
Energy (kJ per day)	9091 (1721)	9394 (2535)	8605 (2515)	7766 (3550)	0.79	0.051	0.25
Protein (%TE)	15 (2.3)	14 (2.5)	15 (4.3)	16 (5.8)	0.71	0.49	0.15
Fat (%TE)	34 (5.9)	36 (5.5)	34 (7.5)	39 (8.0)	0.055	0.65	0.26
Saturated FAs (%TE)	14 (2.9)	14 (2.6)	13 (3.9)	14 (3.8)	0.35	0.38	0.28
Monounsaturated FAs (%TE)	12 (2.4)	12 (2.7)	12 (3.8)	13 (4.5)	0.15	0.59	0.54
*n*-6 polyunsaturated FAs (%TE)	3.9 (1.5)	4.4 (2.0)	3.7 (1.70)	4.5 (3.07)	0.23	0.82	0.83
*n*-3 polyunsaturated FAs (%TE)	0.69 (0.50)	0.58 (0.25)	0.72 (0.51)	0.71 (0.57)	0.58	0.50	0.62
Trans FAs (%TE)	1.05 (0.28)	1.12 (0.27)	1.08 (0.62)	1.27 (0.62)	0.33	0.37	0.52
Cholesterol (mg/day)	245 (106)	236 (97)	256 (176)	222 (144)	0.54	0.14	0.52
Carbohydrate (%TE)	53 (7.1)	52 (6.6)	54 (9.4)	47 (7.3)	0.063	0.10	0.037
Starch (%TE)	27 (3.5)	28 (5.1)	28 (4.7)	25 (8.8)	0.39	0.45	0.044
Total sugars (%TE)	25 (6.3)	24 (7.5)	26 (9.1)	21 (9.0)	0.16	0.30	0.22
Fibre (g/day)	22 (5.1)	19 (6.4)	19 (8.0)	15 (7.1)	0.21 ^	0.46 ^	0.38^

^#^ Values are mean (SD). ^ Corrected for energy intake. %TE: % of total energy; FAs: fatty acids.

### 3.4. Comparison of Micronutrient Intakes Between Lean and Overweight/Obese Pregnant Women During Pregnancy and in Post-Partum Period (Table 3)

The dietary micronutrient intakes (with and without dietary supplements included) and statistical analysis of group effects (overweight/obese vs. lean), time effects (during pregnancy vs. post-partum period), and group * time interactions are shown in Table 3. As there was a trend of lower energy intakes in the post-partum period, the micronutrient intakes were analysed and corrected for energy intake.

**Table 3 nutrients-17-00550-t003:** Micronutrient intakes (vitamins and minerals) in lean versus overweight/obese women during pregnancy (average of V1, V2, and V3) and post-partum (V4) period ^#^.

	During Pregnancy Lean Versus Overweight/Obese		Post-Partum Lean Versus Overweight/Obese		During Pregnancy vs. Post-Partum Lean Versus Overweight/Obese		
Micronutrient (Per Day)	Average Intake During Pregnancy (av V1, V2, V3) Lean (*n* = 26)	Average Intake During Pregnancy (av V1, V2, V3) Overweight/Obese (*n* = 16)	Post-Partum Intake (V4) Lean (*n* = 25)	Post-partum intake (V4) Overweight/Obese (*n* = 15)	*p* ^ Lean vs. Overweight/Obese	*p* ^ Time	*p* ^ Time × Overweight/Obese
* Dietary Vitamin A (µg) (Carotene and Retinol)	793 (422)	638 (308)	922 (1100)	680 (761)	0.34	0.32	0.87
* Dietary Vitamin B1 (mg) (Thiamine)	2.2 (1.6)	1.9 (0.7)	1.6 (0.4)	1.2 (0.5)	0.27	0.18	0.59
* Dietary Vitamin B2 mg (Riboflavin)	2.0 (0.6)	2.0 (0.8)	1.8 (0.7)	1.3 (0.6	0.86	0.87	0.62
* Dietary Vitamin B3 (mg) (Niacin mg)	8.9 (6.1)	7.2 (3.3)	7.9 (7.0)	9.6 (7.4)	0.76	0.066	0.13
Dietary Vitamin B6 (mg) (Pyridoxine)	2.4 (0.7)	2.2 (0.8)	2.0 (0.8)	1.8 (0.7)	0.66	0.60	0.23
Dietary and Supplement Vitamin B6 (mg)	6.7 (3.8)	7.0 (4.5)	4.6 (4.2)	3.1 (3.7)	0.80	0.029	0.66
Vitamin B9 (µg) (Folate)	323 (105)	275 (92)	286 (137)	237 (106)	0.66	0.23	0.11
Dietary and Supplement Vitamin B9 (ug) (Folate)	538 (164)	694 (385)	434 (229)	317 (220)	0.37	0.089	0.26
Vitamin B12 (µg) (Cobalamin)	5.1 (2.4)	4.8 (2.5)	4.5 (2.9)	4.4 (2.7)	0.99	0.41	0.36
Dietary and Supplement Vitamin B12 (µg) (Cobalamin)	7.9 (3.0)	8.0 (3.6)	6.8 (4.9)	6.2 (6.5)	0.57	0.079	0.59
Vitamin C (mg) (L-ascorbic acid)	204 (102)	134 (68)	155 (86)	150 (123)	0.20	0.56	0.081
Dietary and Supplement Vitamin C (mg) (L-ascorbic acid)	243 (109)	199 (79)	193 (97)	167 (131)	0.32	0.78	0.74
Vitamin D (µg) (Calciferol)	3.9 (2.6)	2.6 (1.3)	3.4 (3.4)	1.6 (1.0)	0.089	0.80	0.70
Dietary and Supplement Vitamin D (µg) (Calciferol)	10 (4.4)	10 (4.0)	7.0 (5.8)	3.6 (4.2)	0.42	0.012	0.24
Vitamin E (mg) (Tocopherols and Tocotrienol)	9.6 (2.5)	11 (4.7)	10 (4.7)	9.6 (4.8)	0.52	0.54	0.22
Dietary and Supplement Vitamin E (mg)	12 (4.6)	15 (6.5)	14 (9.6)	14 (11)	0.52	0.89	0.35
Calcium (mg)	1101 (329)	1119 (505)	893 (320)	754 (240)	0.96	0.13	0.42
Dietary and Supplement Calcium (mg)	1122 (342)	1128 (525)	945 (358)	788 (274)	0.98	0.63	0.31
Iodine (µg)	187 (69)	190 (87)	166 (70)	166 (123)	0.42	0.31	0.15
Dietary and Supplement Iodine (µg)	258 (74)	252 (117)	206 (103)	185 (139)	0.72	0.78	0.28
Iron (mg)	13.2 (3.7)	12.3 (4.3)	12.9 (5.3)	9.3 (4.1)	0.27	0.52	0.98
Dietary and Supplement Iron (mg)	60 (80)	77 (127)	67 (164)	12 (8.4)	0.40	0.53	0.27
Magnesium (mg)	307 (59)	273 (79)	277 (77)	224 (62)	0.13	0.38	0.20
Dietary and Supplement Magnesium (mg)	383 (78)	353 (125)	319 (107)	243 (87)	0.17	0.24	0.23
Phosphorous (mg)	1449 (311)	1434 (430)	1256 (312)	1230 (282)	0.36	0.33	0.027
Dietary and Supplement Phosphorous (mg)	1454 (311)	1446 (436)	1262 (315)	1230 (282)	0.38	0.31	0.030
Potassium (mg)	2201 (451)	2424 (581)	2894 (860)	2500 (907)	0.72	<0.0001	0.97
Dietary and Supplement Potassium (mg)	3276 (653)	2998 (827)	2906 (871)	2501 (907)	0.50	0.31	0.21
Sodium (mg)	2934 (732)	3238 (941)	2560 (1183)	2926 (1343)	0.015	0.70	0.19
Sodium to Potassium Ratio	1.40 (0.43)	1.49 (0.43)	0.93 (0.43	1.25 (0.57)	0.021	0.38	0.036
Dietary and Supplement Sodium to Potassium Ratio	0.93 (0.28)	1.17 (0.35)	0.93 (0.43)	1.25 (0.57)	0.0031	0.043	0.098

^#^ Values are mean (SD). * No supplement data for vitamin A, Thiamine, Riboflavin, and niacin. ^ *p* value corrected for total energy intake. There were no sodium supplements taken by any study participant—hence, no data were shown for dietary and supplemental sodium.

Group effects: Sodium intakes were 10% and 13% higher (*p* = 0.015) in the overweight/obese group compared to the lean group during pregnancy and in the post-partum period, respectively. The sodium to potassium intake ratio (without supplements) was 6% and 26% higher (*p* = 0.021) in the overweight/obese group compared to the lean group during pregnancy and in the post-partum period, respectively. The dietary and supplemental sodium to potassium intake ratio was 21% and 26% higher (*p* = 0.0031) in the overweight/obese group compared to the lean group during pregnancy and in the post-partum period, respectively.

Time effects: Vitamin B6 intake (diet and supplemental) was 31% and 56% lower (*p* = 0.029) in the post-partum period compared to during pregnancy for the lean and overweight/obese groups, respectively. Vitamin D intake (diet and supplemental) was 30% and 64% lower (*p* = 0.012) in the post-partum period compared to during pregnancy for the lean and overweight/obese groups, respectively. Potassium intake (without supplements) was 24% and 3% higher (*p* < 0.0001) in the post-partum period compared to during pregnancy for the lean and overweight/obese groups, respectively. The sodium to potassium ratio (diet and supplemental) was maintained in the lean group but increased by 6% (*p* = 0.043) in the overweight/obese group in the post-partum period compared to during pregnancy.

Group * time interactions: There were three significant interaction terms across time and between the two groups. The dietary phosphorous intake was 13% higher in the post-partum period compared to during pregnancy in the overweight/obese group whereas it was 6% lower in the lean group (*p* = 0.027). The dietary and supplemental phosphorous intake was 13% higher in the post-partum period compared to during pregnancy in the overweight/obese group whereas it was 6% lower in the lean group (*p* = 0.030). The sodium to potassium intake ratio was 10% higher in the post-partum period compared to during pregnancy in the overweight/obese group whereas it was 30% lower in the lean group (*p* = 0.036).

### 3.5. Pregnant Women Meeting Harmonised Average Requirements (H-AR) (Table 4)

There was no difference in the proportion of pregnant women meeting the H-AR between the lean and overweight/obese groups, except for vitamin B6 (diet alone) where there was a significantly higher proportion of women meeting the H-AR in the lean group (100%) compared to the overweight/obese group (81%), *p* = 0.022, and vitamin E (diet and supplemental) where there was a significantly lower proportion of women meeting the H-AR in the lean group (35%) compared to the overweight/obese group (69%), *p* = 0.032 (Table 4).

**Table 4 nutrients-17-00550-t004:** Comparison of number (and %) of lean and overweight/obese women meeting H-AR during pregnancy through diet alone and diet plus supplements.

Micronutrient	Rich Food Source	H-AR for Pregnant Women (19–50 Years of Age)	Number (%) Lean (*n* = 26)	Number (%) Overweight/ Obese (*n* = 16)	Comparison of Lean and Overweight/ Obese *p* Value
* Vitamin A (µg/day) (Carotene and retinol)	Meat, fish, liver, eggs	≥540	17 (65%)	9 (56%)	0.45
* Vitamin B1 (mg/day) (Thiamine)	Liver, pork, fish, legumes	≥1.2	26 (100%)	14 (88%)	0.065
* Vitamin B2 (mg/day) (Riboflavin)	Meat, fortified food	≥1.5	21 (81%)	9 (56%)	0.088
* Vitamin B3 (mg/day) (Niacin mg)	Meat, fish, brown rice	≥14	3 (12%)	0 (0%)	0.16
Vitamin B6 (mg/day) (Pyridoxine)	Meat	≥1.5	26 (100%)	13 (81%)	0.022
Vitamin B6 + supplements (mg/day)			26 (100%)	15 (94%)	0.20
Vitamin B9 (µg/day) (Folate)	Green leafy vegetables	≥520	1 (4%)	0 (0%)	0.43
Folate + supplements (ug/day)			13 (50%)	11 (69%)	0.23
Vitamin B12 (µg/day) (Cobalamin)	Meat	≥2.2	23 (88%)	16 (100%)	0.16
Vitamin B12 + supplements (µg/day)			26 (100%)	16 (100%)	n/a
Vitamin C (mg/day) (L-ascorbic acid)	Citrus fruit	≥80	22 (85%)	12 (75%)	0.44
Vitamin C + supplements			25 (96%)	15 (94%)	0.72
Vitamin D (µg/day) (Calciferol)	Fatty fish, eggs	≥10	1 (4%)	0 (0%)	0.43
Vitamin D + supplements (µg/day)			14 (54%)	9 (56%)	0.88
Vitamin E (mg/day) (Tocopherols and Tocotrienol)	Green leafy vegetables, wholegrains	≥12	6 (23%)	8 (50%)	0.072
Vitamin E + supplements (mg/day)			9 (35%)	11 (69%)	0.032
Calcium (mg/day)	Dairy products	≥860 (19–30 yrs) ≥750 31–50 yrs)	21 (81%)	12 (75%)	0.66
Calcium + supplements (mg/day)			21 (81%)	12 (75%)	0.66
Iodine (µg/day)	Fish/seafood, seaweed, iodised salt	≥160	14 (54%)	10 (63%)	0.58
Iodine + supplements (µg/day)			24 (92%)	14 (88%)	0.61
Iron (mg/day)	Meat, spinach	≥11.2 ^	19 (73%)	9 (56%)	0.26
Iron + supplements (mg/day)			26 (100%)	15 (94%)	0.20
Magnesium (mg/day)	Green leafy vegetables, nuts, seeds, wholegrains	≥290 (19–30 yrs) ≥300 (31–50 yrs)	13 (50%)	7 (44%)	0.69
Magnesium + supplements (mg/day)			23 (88%)	11 (69%)	0.11
Phosphorous (mg/day)	Dairy, meat, legumes	≥580	26 (100%)	16 (100%)	n/a
Phosphorous + supplements (mg/day)			26 (100%)	16 (100%)	n/a
Potassium (mg/day)	Fruit and vegetables	≥3500 ^#^	26 (100%)	16 (100%)	n/a
Potassium + supplements (mg/day)			26 (100%)	16 (100%)	n/a
Sodium (mg/day)	Processed food	<2400 ^#^	20 (77%)	13 (79%)	0.74
Sodium + supplements (mg/day)			20 (77%)	13 (79%)	0.74

* No supplement data for vitamin A, Thiamine, Riboflavin, and niacin. ^ For pregnant women, H-AR for iron is 7, 11.2, or 22.4 mg/day corresponding to high, moderate, or low iron absorption, respectively [8]. ^#^ There is no H-AR for potassium and sodium; therefore, comparison was made to UK DRV NRI [5]. n/a = 100% meeting H-AR for both groups.

Of the 17 micronutrients analysed without contributions from dietary supplements, all women met the H-AR for only two micronutrients: phosphorous and potassium (Table 4). Deficiencies were notable for niacin, folate, and vitamin D as more than 77%, 95%, and 95% of women, respectively, did not satisfy the H-AR dietary requirements with the consumption of foods only. The number of deficiencies were reduced for folate and vitamin D with supplements, but the deficiencies remained high in a significant proportion of women. Deficiencies in the other micronutrients were not as severe but still notable for vitamin A (35% in lean, 44% in overweight/obese); Riboflavin (44% overweight/obese); vitamin E (77% lean, 50% overweight/obese); iodine (46% lean, 37% overweight/obese); iron (44% overweight/obese); and magnesium (50% lean, 56% overweight/obese). The number of deficiencies in these nutrients were improved with supplements; however, a large proportion of women still did not meet the H-AR for vitamin E (65% lean, 31% overweight/obese) and magnesium (31% overweight/obese) (Table 4).

Dietary supplements increased the proportion of pregnant women meeting the H-AR (Table 4), notably for folate, vitamin D, iodine, iron, magnesium, and vitamin E, which reached greater than 50% H-AR (except for vitamin E in the lean group) when supplements were taken.

### 3.6. Offspring Outcomes

The weight of the offspring did not significantly differ between the two groups (3.3 ± 0.5 versus 3.6 ± 0.6 kg for lean and overweight/obese groups, respectively; *p* = 0.25). The birth weight centile also did not significantly differ (45.5 ± 26.6 versus 39.3 ± 29.7 for lean and overweight/obese groups, respectively; *p* = 0.50).

## 4. Discussion

The principal findings were that there were no differences in macronutrient intakes between overweight/obese and lean women during pregnancy, and overweight/obese women consumed higher amounts of sodium and had a higher sodium to potassium ratio compared to the lean group. More than 87% of all women did not meet the H-AR for niacin; more than 95% of women did not meet the H-AR for folate and vitamin D through diet alone.

The energy intakes from fat and carbohydrate but not protein in the Scottish women were slightly higher than the mean of the 54 studies reported in the meta-analysis of maternal macronutrient and energy intake during pregnancy [20], but there were no significant differences in energy and macronutrient intakes between the overweight/obese and lean pregnant women groups in the current study.

The increased sodium intakes and sodium to potassium ratio in the overweight/obese women compared to the lean group would suggest that the overweight/obese women consumed more processed foods [28] and less unprocessed, wholesome foods, like fruit and vegetables, given that there is a correlation between potassium intakes and fruit and vegetable intakes [29]. However, in this study, there were no differences in the intakes of sodium-rich foods and fruit and vegetable intake between the two groups, likely due to the small sample size.

Micronutrients, such as niacin, folate, and vitamin D, were consumed in very low amounts and more than 87%, 95%, and 95% of women did not meet the H-AR for niacin, folate, and vitamin D, respectively, from diet alone. Furthermore, >76% of women from the lean group and >55% of women from the overweight/obese group did not meet the H-AR for vitamin E and magnesium, respectively. This shows that a large proportion of pregnant women are not meeting micronutrient intakes required for optimal health. Other studies have reported similar/mixed results. In Saudi Arabia, the vitamin D intake was 40 IU (approximately 2 µg per day) in pregnant women [30], which was much lower than the current study. Iron intakes in the USA were 14.4 mg per day [18] and 18.7 mg per day in Uganda [31], which were both higher than the current study. Folate intakes were very low in Uganda, ranging from 121 to 310 µg per day, which translated to 0–7% of pregnant women meeting the 600 µg per day recommended intakes [31], whereas folate intake (dietary intake alone) was 432 µg per day in the USA [18], which was higher than the current study, most likely due to folate enrichment of flour in the USA since 1996 [32]. Calcium intake in the USA was 1033 mg per day [18], similar to the current study, but calcium intake in China was only 602 mg per day [33], likely due to low milk consumption (China’s consumption is ranked at 137 out of 177 countries compared to the UK being ranked at 24) with actual consumption being 333 kg/capita/year in 2013 compared to 232 kg/capita per year in the UK in 2013 (https://en.wikipedia.org/wiki/List_of_countries_by_milk_consumption_per_capita, accessed on 18 December 2024). Calcium intake was also low in Uganda, ranging from 312 to 1018 mg per day depending on region and whether during planting or harvesting season [31] and this translated to 0–55% of pregnant women meeting the recommended intakes of 1000 mg per day for calcium. Iron intakes in Uganda ranged from 9.9 to 22.4 mg per day, resulting in 0–55% of pregnant women meeting the recommended intakes of 27 mg per day, whereas iron intake in the USA [18] and China [33] was 14.4 and 18.7 mg per day, respectively, which were higher than the current study. Collectively, these few studies on micronutrient intakes during pregnancy highlight that some micronutrient intakes do not meet the recommended dietary intakes for pregnant women, which may have detrimental consequences for the growing foetus.

Folate is essential for the prevention of NTDs, as shown by the definitive MRC study [6], and several other micronutrients, including niacin, are associated with a reduced risk of NTDs [34]. Less than 5% of pregnant women met the 520 µg H-AR for folate during pregnancy through diet alone; hence, many women relied on folate supplementation to meet the H-AR. Green leafy vegetables are a rich source of dietary folate, but it was reported that pregnant women have low vegetable intake [10]. Unlike folate, niacin is not an essential nutrient as humans can synthesise niacin from the amino acid tryptophan, where it is estimated that 60 mg of dietary tryptophan is equivalent to 1 mg niacin [35].

Vitamin D helps to maintain muscle and bone strength and helps the body to absorb calcium from food. In pregnancy, vitamin D also aids bone development and the amount of calcium within the baby’s bones can be affected by a vitamin D deficiency in the mother. For this reason, there are specific recommendations for vitamin D supplementation during pregnancy [36]. Vitamin D supplementation was shown to be safe up to 4000 IU and is largely devoid of any adverse pregnancy outcomes [15]. Moreover, it was shown that maternal vitamin D intake affects postnatal growth and is inversely associated with childhood overweight in children of mothers with normal weight [37], highlighting the importance of pre-pregnancy weight status in addition to its association with diet quality [38]. Vitamin D can be synthesised by the body through sunlight exposure and 9 minutes of daily exposure to sunlight will maintain vitamin D status in the UK through their winter [39], but this is dependent on sunlight.

Vitamin E was found to be very effective in the prevention and reversal of various disease complications due to its function as an antioxidant, its role in anti-inflammatory processes, its inhibition of platelet aggregation, and its immune-enhancing activity [40]. Vitamin E intake was associated with several health benefits (coronary heart disease, cancer, eye disorders, and cognitive decline); however, supplementing the diet with vitamin E is not supported by scientific evidence [40] and at present, there is no UK recommended intake for pregnant women.

Additionally, iron is needed for erythrocyte synthesis to support an increase in blood volume during pregnancy and to prevent anaemia. The WHO key facts state that globally 37% of pregnant women are affected by anaemia [41]. Therefore, pregnant women who are micronutrient deficient during pregnancy are putting themselves and their babies at risk of suboptimal health and birth defects.

There are plenty of resources for nutrition advice in pregnancy including from the UK [1,42], Australia [43] and the USA [9]. Nevertheless, given the current results, there is a need for health care providers to provide a simple message such as to ‘eat better and not more’ [9], as well as specific advice on foods that are rich in niacin, folate, vitamin D, and vitamin E. Therefore, the advice should recommend the consumption of fish (preferably oily), meats (including lean red meat and chicken/turkey breast meat), and unprocessed foods including green leafy vegetables, legumes (including peas and beans), and nuts, as well as providing specific advice to refrain from consuming foods rich in salt, including ultra-processed foods. An example of such a diet is the Mediterranean diet, which provides numerous health benefits including improved cardiovascular health [44].

The strength of this study is the longitudinal repeated measures design during pregnancy and the average of actual intakes during pregnancy. The limitation is that this study had a small sample size; however, notable statistical differences were detected.

## 5. Conclusions

Overweight/obese pregnant women consumed more sodium and had a higher sodium to potassium ratio compared to the lean group. A large proportion of pregnant women, irrespective of BMI, were not meeting the H-AR for micronutrient intakes, notably niacin, folate, vitamin D, magnesium, and vitamin E and therefore a simple message of ‘eat better and not more’ is supported.

## Figures and Tables

**Table 1 nutrients-17-00550-t001:** Maternal study participants’ characteristics ^#^.

	Lean (*n* = 26)	Overweight/Obese (*n* = 16)	*p* Value
Age (years)	30.1 (2.9)	30.8 (4.7)	0.61
Education level (*n*, %)			
HNC	0 (0%)	2 (13%)	0.19
HND	1 (4%)	2 (13%)	
Secondary	2 (8%)	1 (6%)	
University	23 (88%)	11 (69%)	
SIMD (*n*,%)			
1	2 (8%)	7 (44%)	0.069
2	5 (19%)	2 (13%)	
3	4 (15%)	3 (19%)	
4	3 (12%)	1 (6%)	
5	12 (46%)	3 (19%)	
Alcohol (units per week) (*n*,%)			
0	23 (88%)	16 (100%)	0.37
0.5	1 (4%)	0 (0%)	
1.0	2 (8%)	0 (0%)	
Height (m)	1.66 (0.07)	1.67 (0.06)	
Weight (kg)			
V1	60.7 (7.2)	87.0 (10.2)	<0.0001
V2	65.5 (7.0)	91.7 (10.3)	
V3	71.1 (7.0)	96.5 (10.6)	
V4	62.8 (7.0) *	86.8 (9.8)	
BMI (kg/m^2^)			
V1	22.1 (1.9)	31.1 (2.9)	<0.0001
V2	23.9 (2.0)	32.8 (2.7)	
V3	25.9 (2.1)	34.5 (2.5)	
V4	22.8 (1.8) *	31.1 (3.1)	
Systolic blood pressure (mmHg)	110 (9)	119 (10)	0.014
Diastolic blood pressure (mmHg)	68 (6)	74 (8)	0.016
Gestation (weeks)			
V1	15.7 (1.1)	16.0 (1.5)	0.51
V2	25.0 (1.3)	25.3 (1.1)	
V3	35.5 (1.1)	35.4 (1.0)	
Post-partum (weeks)	12.6 (1.6)	12.2 (1.2)*	0.40

^#^ Values are mean (SD) unless otherwise specified. *n* = number. HNC—Higher National Certificate; HND—Higher National Diploma; V1—first trimester; V2—s trimester; V3—third trimester; V4—post-partum. BMI—Body Mass Index. SIMD—Scottish Index of Multiple Deprivation. SIMD looks at extent to which an area is deprived across seven domains: income, employment, education, health, access to services, crime, and housing. SIMD is 5-point scale where 1 is most deprived. * Missing *n* = 1.

## Data Availability

Data will be made available upon reasonable request to the corresponding author. The data are not publicly available due to ethical reasons.

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
