# Peer review of "Dietary Micronutrient Intake During Pregnancy Is Suboptimal in a Group of Healthy Scottish Women, Irrespective of Maternal Body Mass Index"

_nutrients, 2025, doi:10.3390/nu17030550_

Round 1
Reviewer 1 Report
Comments and Suggestions for Authors
Dear colleagues,
I reviewed with interest your study. It compared the nutrient intakes between Scottish lean and overweight/obese women during pregnancy, and which interesting information. The study was done relatively many years ago (2010 to 2011). Nevertheless, the data may still be worthy to share. However, I noted that you have a different number of individuals in each group: 26 and 16, respectively. I assume that you may have excluded 10 overweight/obese women that developed hypertension or gestational diabetes throughout the study. In my opinion, this information is essential to make a fair and meaningful comparison between the two groups. Otherwise, you are selecting a “healthy” overweight/obese group, when indeed it is important, for this study, to identify the dietary differences.
You also used values of Reference Nutrient Intakes (RNI) to make the comparisons, when this study was comparing two “population” group, and therefore you should have used Average Requirements (AR), which are lower than the RNI’s. The latter are used for studying diets of individuals and not populations. Furthermore, you increased the folate dietary requirement during pregnancy, when the higher recommendations (400-600 ug/day) are for preventing neural tube defects before pregnancy. Therefore, there is no justification to use this value for your study. Similarly, the dietary recommendation of vitamin D depends on the sun exposure of the population. You may discuss this situation in your manuscript. I would suggest that you consult the following references for deciding the AR values that you may use in a future manuscript: (1) EFSA Dietary Reference Values for Nutrients, updated 2019: doi: 10.2903/sp.efsa.2017.e15121; (2) Allen LH, Carriquiry AL, Murphy SP. Perspective: Proposed harmonized nutrient reference values for populations. Adv Nutr 2020; 11:469-483. doi: https://doi.org/10.1093/advances/nmz096.
In summary, I am recommending to reconsidering the analysis of your data following the suggestions that I have presented
Some minor suggestions are as follows:
1. Switch table 2 with the table in the annex. It is preferable to discuss with priority absolute intakes of the macronutrients and not their proportional composition in the diet (the actual table 2, which may go as an Annex).
2. Split table 1 in two: (1) anthropometric parameters and gestation (weeks); and (2) the rest of the groups’ characterization.
3. Table 3 (as well as Table 2): Make the statistic comparisons between lean and overweight/obese groups during pregnancy and postpartum, separately; and then each one of these groups independently during pregnancy and post-partum. Moreover, write SD besides the mean and not below it.
4. Table 4: Repeat including all data from the overweight/obese group and using as the adequacy references the Average Requirement (either the EFSA or the international harmonized recommendations).
Author Response
Reviewer 1
Dear colleagues,
I reviewed with interest your study. It compared the nutrient intakes between Scottish lean and overweight/obese women during pregnancy, and which interesting information. The study was done relatively many years ago (2010 to 2011). Nevertheless, the data may still be worthy to share. However, I noted that you have a different number of individuals in each group: 26 and 16, respectively. I assume that you may have excluded 10 overweight/obese women that developed hypertension or gestational diabetes throughout the study. In my opinion, this information is essential to make a fair and meaningful comparison between the two groups. Otherwise, you are selecting a “healthy” overweight/obese group, when indeed it is important, for this study, to identify the dietary differences.
Response: We agree with the reviewer that the data for this study was collected many years ago but that it is still relevant to publish today.
We have added information regarding the power calculation to section 2.4 Statistical Analysis.
The study numbers differed between the two groups primarily due to more women with a healthy body weight consenting to participate than those women who were overweight/obese. We have added the following to the results section 3.1 the following has been added to the results section 3.1 “Thirty lean and 22 overweight/obese women were recruited at the first antenatal clinic appointment to take part in the study and 26 lean and 16 overweight/obese women completed the study.”
The aim of our study was to compare any differences in intakes between women who were overweight/obese, who were devoid of any metabolic disease (e.g. GDM and/or hypertension), compared to women who had a healthy body weight. The reason for the exclusion of any metabolic disease, as we wanted to see the potential differences in intakes due to body weight differences alone and not due to any metabolic diseases. The aim of the study has been clarified to reflect this.
You also used values of Reference Nutrient Intakes (RNI) to make the comparisons, when this study was comparing two “population” group, and therefore you should have used Average Requirements (AR), which are lower than the RNI’s. The latter are used for studying diets of individuals and not populations. Furthermore, you increased the folate dietary requirement during pregnancy, when the higher recommendations (400-600 ug/day) are for preventing neural tube defects before pregnancy. Therefore, there is no justification to use this value for your study. Similarly, the dietary recommendation of vitamin D depends on the sun exposure of the population. You may discuss this situation in your manuscript. I would suggest that you consult the following references for deciding the AR values that you may use in a future manuscript: (1) EFSA Dietary Reference Values for Nutrients, updated 2019: doi: 10.2903/sp.efsa.2017.e15121; (2) Allen LH, Carriquiry AL, Murphy SP. Perspective: Proposed harmonized nutrient reference values for populations. Adv Nutr 2020; 11:469-483. doi: https://doi.org/10.1093/advances/nmz096.
Response: we thank the reviewer for pointing this out and we have used the suggested reference by Allen et al and compared our women’s micronutrients intakes to those recommended for pregnant women by Allen et al (2020). We have updated the manuscript throughout accordingly. We have also added a statement re: vitamin D and sunlight exposure to the discussion section.
In summary, I am recommending to reconsidering the analysis of your data following the suggestions that I have presented
Some minor suggestions are as follows:
- Switch table 2 with the table in the annex. It is preferable to discuss with priority absolute intakes of the macronutrients and not their proportional composition in the diet (the actual table 2, which may go as an Annex).
Response: we thank the reviewer for this suggestion; however we disagree with this suggestion. Even though there were no statistically significant differences in energy intakes between the two groups (Table 2, p=0.051), we wanted to see if there were any differences in intakes irrespective of the differences in energy intake, hence we expressed the data as percent of total energy intake.
We also felt it necessary to report the actual intakes and therefore we added the actual intakes in the Appendix.
- Split table 1 in two: (1) anthropometric parameters and gestation (weeks); and (2) the rest of the groups’ characterization.
Response: we thank the reviewer for this suggestion, however, reviewer 2 has suggested that we transpose the table such that Lean and Overweight/obese become the column headings and the characteristics are listed in the first column. Therefore, we have changed the presentation of Table 1 as suggested by the other reviewer.
- Table 3 (as well as Table 2): Make the statistic comparisons between lean and overweight/obese groups during pregnancy and postpartum, separately; and then each one of these groups independently during pregnancy and post-partum. Moreover, write SD besides the mean and not below it.
Response: we thank the reviewer for this suggestion, however we disagree that we should analyse the data separately as this is a longitudinal study with repeated measures in the same women. Therefore, it would be erroneous to separate out the statistical analysis as suggested and leave us open to multiple testing and Type 1 error. We used the mixed model analysis to avoid this.
We have written the SD next to mean as suggested by the reviewer.
- Table 4: Repeat including all data from the overweight/obese group and using as the adequacy references the Average Requirement (either the EFSA or the international harmonized recommendations).
Response: This has been addressed and Table 4 has been updated. We have updated the manuscript throughout accordingly.
Reviewer 2 Report
Comments and Suggestions for Authors
The paper is well-written and presents an interesting idea. However, there are several aspects that need to be addressed before it can be considered for publication. Due to the limitations in sample size and participant selection criteria, the title should be revised to better reflect the content of the study.
Methods:
Line 126: The author mentions the "multiple pass method" and its wide implementation in various demographic study groups. Could you please provide citations to support this statement? Additionally, could you kindly clarify what the "multiple pass method" entails for the readers who may not be familiar with it?
Statistical Power: Did the author consider the statistical power in relation to the number of participants included in the study, especially given that the survey spans multiple time points? Clarification on this would help readers better understand the robustness of the study's findings.
Gestational Weight Gain: Since the participants were classified as lean and overweight/obese, were the doctors involved in the study advising them to manage their gestational weight gain? Could this guidance potentially influence their dietary intake or overall behavior during the study?
Exclusion Criteria: The authors have excluded participants with preexisting metabolic diseases, gestational diabetes, or hypertension disorders. Does this suggest that the overweight/obese participants in the study were relatively healthy despite their BMI classification? Could this have any impact on the interpretation of the results?
SIMD Classification: There is no description of the SIMD (Scottish Index of Multiple Deprivation) in the manuscript. Based on the data, it appears that most overweight and obese participants belong to SIMD 1 and 2. Would it be possible to combine these into two categories (<3, ≥3) for clearer analysis? A brief explanation would also be helpful for readers unfamiliar with the SIMD scale.
Confounders: Without adjusting for potential confounders, the conclusions might need to be presented with more caution. It would be beneficial to address this concern or clarify whether confounder variables were considered in the analysis.
Health Insurance Impact: Could the health insurance system have had any impact on the participants' choice of supplementation or other dietary interventions? A brief discussion of this factor might add valuable context to the findings.
Results:
Table 1: It would be helpful to clarify the timing of each measurement in Table 1 or provide definitions for each measure to distinguish them more clearly. This would improve the understanding of the data presented.
Table 1 Transposition: It may be more effective to transpose Table 1, as many fields are left blank in the current layout. For example, presenting the data in the following manner might help:
|
  |
lean |
overweight/obese |
|
Maternal Age |
||
|
Educational level |
||
|
SIMD |
||
|
alcohol |
||
|
Height |
||
|
weight v1 |
||
|
weight v2 |
||
|
weight v3 |
||
|
weight v4 |
||
|
BMI V1 |
||
|
BMI V2 |
||
|
BMI V3 |
||
|
BMI V4 |
  |
  |
I’ve started this for reference, but it would be ideal if the table could be fully transposed for consistency and clarity.
Table 2: The sum of protein (%TE), fat (%TE), and carbohydrate (%TE) exceeds 100%. Could the authors please verify and explain this discrepancy? A clarification would help in interpreting the nutritional data more accurately.
Table 2 - Energy Intake: In Table 2, it is observed that overweight/obese pregnant women had higher energy intake during pregnancy compared to lean pregnant women, yet their intake was much lower during the postpartum period. Could the authors elaborate on the possible factors that contributed to this difference? An explanation would be valuable to understand the underlying reasons behind this trend.
Author Response
Reviewer 2
The paper is well-written and presents an interesting idea. However, there are several aspects that need to be addressed before it can be considered for publication. Due to the limitations in sample size and participant selection criteria, the title should be revised to better reflect the content of the study.
Response: we agree with the reviewer and the title has been changed to “Dietary micronutrient intake during pregnancy is suboptimal in a group of healthy Scottish women, irrespective of maternal body mass index”
Methods:
Line 126: The author mentions the "multiple pass method" and its wide implementation in various demographic study groups. Could you please provide citations to support this statement? Additionally, could you kindly clarify what the "multiple pass method" entails for the readers who may not be familiar with it?
The publication by “Conway, J.M.; Ingwersen, L.A.; Moshfegh, A.J. Accuracy of dietary recall using the USDA five-step multiple-pass method in men: an observational validation study. J. Am. Diet. Assoc. 2004, 104, 595-603, doi:10.1016/j.jada.2004.01.007”, has been cited 506 times (see link: https://www.semanticscholar.org/paper/Accuracy-of-dietary-recall-using-the-USDA-five-step-Conway-Ingwersen/da7db3738cf094b68147ee5c74a12d77aeaaf215) and is included in the reference list.
The multiple-pass method is a five-step process for collecting dietary data that involves multiple passes through a 24-hour period:
- Quick list: The subject lists all foods and beverages consumed without interruption
- Forgotten foods: The subject is asked about foods they may have forgotten to list
- Time and occasion: The subject records the time and occasion for each food
- Detail cycle: The subject provides detailed information about each food, including the amount consumed, cooking methods, and brands
- Final probe: The subject is asked if they consumed anything else
This description has been added to the methods section.
Statistical Power: Did the author consider the statistical power in relation to the number of participants included in the study, especially given that the survey spans multiple time points? Clarification on this would help readers better understand the robustness of the study's findings.
The study participants completed the multiple pass 24-hour dietary recall four times in total, three times during pregnancy (i.e. trimester 1, 2 and 3) and one time during the post-partum period. Given the limitations of 24-hour dietary recall, the authors decided to take an average of the three 24-hour dietary recalls during pregnancy as this would be more representative of their true dietary intake.
We have now clarified this in the methods section (section 2.2), where the original statement of “An average of V1, V2 and V3 dietary intakes were calculated to obtain a more representative dietary intake during pregnancy” now reads “Given the limitations of the multiple pass 24-hour dietary recall, an average of V1, V2 and V3 dietary intakes were calculated to obtain a more representative dietary intake during pregnancy”.
We have added information regarding the power calculation to section 2.4 Statistical Analysis “For the Lipotoxicity in Pregnancy Study [22], the numbers recruited were based on power calculations for differences in the anthropometric measures [24], energy expenditure [25], endothelial function [26] and lipotoxic measures [27] but not for biological assays. The number of participants required was worked out was based on the mean obese measurement of the described parameter minus the mean lean measurement divided by the mean lean. This is described at the standardised delta. The second calculation required for the power was the sigma of the lean group divided by the mean lean. These results were then exported to Minitab vs16 where numbers needed to recruit were based on a 2 sample t power calculation. As the largest number needed to recruit was based on endothelial function (n=24), it was decided to aim to recruit 30 women to each BMI group in order to detect a difference and cover any study drop outs.”
Moreover, the following has been added to the results section 3.1 “Thirty lean and 22 overweight/obese women were recruited at the first antenatal clinic appointment to take part in the study and 26 lean and 16 overweight/obese women completed the study.”
The 4 additional references have been added to the reference list.
[24] Soltani, H.; Fraser, R. B. A longitudinal study of maternal anthropometric changes in normal weight, overweight and obese women during pregnancy and postpartum. Br. J. Nutr, 2000, 84, 95-101.
[25] Catalano P. M.; Roman-Drago, N. M.; Amini, S. B.; Sims, E. A. Longitudinal changes in body composition and energy balance in lean women with normal and abnormal glucose tolerance during pregnancy. Am. J. Obstet. Gynecol. 1998, 179, 156-165.
[26] Stewart, F.; Freeman, D. J.; Ramsay, J. E.; Greer, I. A.; Caslake, M.; Ferrell, W. R. Longitudinal assessment of maternal endothelial function and markers of inflammation and placental function throughout pregnancy in lean and obese mothers. J. Clin. Endocrinol. Metab, 2007, 92, 969-975.
[27] Stewart, F. M. The impact of maternal obesity on vascular and metabolic function throughout pregnancy. 2007. MD, University of Glasgow.
Gestational Weight Gain: Since the participants were classified as lean and overweight/obese, were the doctors involved in the study advising them to manage their gestational weight gain? Could this guidance potentially influence their dietary intake or overall behavior during the study?
Response: The study participants were under the general antenatal care and there was no specific gestational weight gain advice, and they were given no additional specific dietary advice.
Exclusion Criteria: The authors have excluded participants with preexisting metabolic diseases, gestational diabetes, or hypertension disorders. Does this suggest that the overweight/obese participants in the study were relatively healthy despite their BMI classification? Could this have any impact on the interpretation of the results?
Yes we excluded study participants that had preexisting metabolic diseases and therefore the overweight/obese women are relatively healthy despite their BMI classification. The reason for the exclusion of any metabolic disease, as we wanted to see the potential differences in dietary intakes due to body weight differences alone and not due to any metabolic diseases. The aim of the study has been clarified to reflect this. No, this exclusion would not have any impact on the interpretation of the results in the context of healthy pregnancies, not to the general maternity population.
SIMD Classification: There is no description of the SIMD (Scottish Index of Multiple Deprivation) in the manuscript. Based on the data, it appears that most overweight and obese participants belong to SIMD 1 and 2. Would it be possible to combine these into two categories (<3, ≥3) for clearer analysis? A brief explanation would also be helpful for readers unfamiliar with the SIMD scale.
Response: The re-analysis of SIMD as <3 and ≥ 3 showed that 7 lean (27%) and 9 overweight/obese (56%) women had scores of less than 3, with a p value of 0.057. As there is no statistical significance in SIMD, we have not collapsed the categories in Table 1.
SIMD is only mentioned in Table 1 itself and in the footnote of table 1 where to states “SIMD – Scottish Index of Multiple Deprivation, where 1 is most deprived”, which has been explained as follows “SIMD – Scottish Index of Multiple Deprivation. SIMD looks at the extent to which an area is deprived across seven domains: income, employment, education, health, access to services, crime and housing The SIMD is a 5 points scale where 1 is most deprived.” https://www.gov.scot/collections/scottish-index-of-multiple-deprivation-2020/
Due to lack of statistical significance SIMD is only mentioned in Table 1 and its footnote and not in the manuscript.
Confounders: Without adjusting for potential confounders, the conclusions might need to be presented with more caution. It would be beneficial to address this concern or clarify whether confounder variables were considered in the analysis.
Response: There were no confounding variables identified, i.e. there were no smokers, no significant differences in education level, socioeconomic status (SIMD), alcohol consumption or gestation weeks and therefore they were not considered in the analysis.
Health Insurance Impact: Could the health insurance system have had any impact on the participants' choice of supplementation or other dietary interventions? A brief discussion of this factor might add valuable context to the findings.
We thank the reviewer for your suggestion. All participants were undergoing care in the National Health Service (NHS) in Scotland which is a universal health care service free at the point of delivery. All drug prescriptions in Scotland are free. The NHS in Scotland provides free vitamins to anyone who requires them during pregnancy, ‘health start’ vitamins, or women can choose the supplements they want and buy them. Health Insurance is not an issue in the UK.
Results:
Table 1: It would be helpful to clarify the timing of each measurement in Table 1 or provide definitions for each measure to distinguish them more clearly. This would improve the understanding of the data presented.
Table 1 Transposition: It may be more effective to transpose Table 1, as many fields are left blank in the current layout. For example, presenting the data in the following manner might help:
|
  |
lean |
overweight/obese |
|
Maternal Age |
|
|
|
Educational level |
|
|
|
SIMD |
|
|
|
alcohol |
|
|
|
Height |
|
|
|
weight v1 |
|
|
|
weight v2 |
|
|
|
weight v3 |
|
|
|
weight v4 |
|
|
|
BMI V1 |
|
|
|
BMI V2 |
|
|
|
BMI V3 |
|
|
|
BMI V4 |
  |
  |
I’ve started this for reference, but it would be ideal if the table could be fully transposed for consistency and clarity.
Response: We thank the reviewer for this suggestion and we have changed the presentation of Table 1 as suggested.
Table 2: The sum of protein (%TE), fat (%TE), and carbohydrate (%TE) exceeds 100%. Could the authors please verify and explain this discrepancy? A clarification would help in interpreting the nutritional data more accurately.
The sum of of protein (%TE), fat (%TE), and carbohydrate (%TE) exceeds 100% slightly (i.e.102%) and this can be explained by rounding to whole numbers during the calculations to express as percent of total energy and average intakes. Note that this rounding effect is consistent across all groups.
Table 2 - Energy Intake: In Table 2, it is observed that overweight/obese pregnant women had higher energy intake during pregnancy compared to lean pregnant women, yet their intake was much lower during the postpartum period. Could the authors elaborate on the possible factors that contributed to this difference? An explanation would be valuable to understand the underlying reasons behind this trend.
We thank the reviewer for their comment, however, there were no statistical significant differences in energy intake during pregnancy between the two groups (p=0.79), or any statistical significant differences across time (p=0.051) or the interaction term (p=0.25). The closest result to being significant, were that both groups reduced their energy intake in the post-partum period compared to during pregnancy (p=0.051).
Given that there were no statistical significant results to discuss, we have not elaborated on this non-significant result.
Furthermore, to confirm that there were no statistical significant differences between energy intake in the post-partum period between the two groups using a t-test resulted in a p=0.47 which is not statistically significant.
Round 2
Reviewer 1 Report
Comments and Suggestions for Authors
Dear colleagues,
Thanks for your clarifications regarding why the study did not include the same number of women in the two groups. I also noticed that you disagree with some of my initial suggestions, and I accept your position. Finally, thanks for accepting some of my recommendations to use different dietary reference intakes (DRI) to interpret your results for populations. In general, your manuscript is now clearer.
I would like to recommend to improve the paragraph between lines 151 and 154. The UK DRV values, although deduced from population studies, as they are based on the 97% percentile of requirement, they are applicable for the assessment of diets of individuals. On the contrary, the H-AR values are used for studying diets of populations or comparing groups as you did in this study. Please make those corrections in your text.
I would also like to recommend to modify the description of the results of Table 4 as presented in lines 283-288 to emphasize micronutrient inadequacies. Something like this may work: Inadequacy was serious for niacin, folate, and vitamin D, as 74%, 96%, 96%, or more of women of the two groups did not satisfy the dietary requirement only with foods. Inadequacy was reduced for folate and vitamin D with supplements, but inadequacy remained high in an important proportion of women. Inadequacy for the other micronutrients was inexistent or mild (combining diet and supplementation), although it was still important for vitamin A, vitamin E, and vitamin B2. Lean women with inadequacy were 35% and 65% for vitamin A and E, respectively. And, for overweight/obese women the inadequacy was identified in 44% and 31% for these micronutrients, and 44% for vitamin B2 (Table 4)”. I am certain that you can improve the grammar of these paragraph but my point is to focus on the inadequacy rates.
Lines 302 and 303, and lines 316-319, of the discussion should also b modified to match with the description of the dietary inadequacies as mentioned above.
Finally, similarly to the comment that vitamin D can be synthesized through exposure of the skin to sunlight, you can add a sentence saying that niacin may be synthesized using the amino-acid tryptophan as a precursor.
Author Response
Dear colleagues,
Thanks for your clarifications regarding why the study did not include the same number of women in the two groups. I also noticed that you disagree with some of my initial suggestions, and I accept your position. Finally, thanks for accepting some of my recommendations to use different dietary reference intakes (DRI) to interpret your results for populations. In general, your manuscript is now clearer.
I would like to recommend to improve the paragraph between lines 151 and 154. The UK DRV values, although deduced from population studies, as they are based on the 97% percentile of requirement, they are applicable for the assessment of diets of individuals. On the contrary, the H-AR values are used for studying diets of populations or comparing groups as you did in this study. Please make those corrections in your text.
Response:
We have replaced “The UK DRV are based on population intakes that would be sufficient for 97% of the population [5], whilst the H-AR are used to assess diets of individuals and not populations [8]” with “The UK DRV are estimates of the daily amounts of nutrients that healthy people need. They are used to assess the safety and adequacy of nutrient intake in a population [5]. The H-AR is a nutrient intake level that meets the needs of half of a population. It is used to estimate how many people in a group are deficient in a nutrient [8].”
I would also like to recommend to modify the description of the results of Table 4 as presented in lines 283-288 to emphasize micronutrient inadequacies. Something like this may work: Inadequacy was serious for niacin, folate, and vitamin D, as 74%, 96%, 96%, or more of women of the two groups did not satisfy the dietary requirement only with foods. Inadequacy was reduced for folate and vitamin D with supplements, but inadequacy remained high in an important proportion of women. Inadequacy for the other micronutrients was inexistent or mild (combining diet and supplementation), although it was still important for vitamin A, vitamin E, and vitamin B2. Lean women with inadequacy were 35% and 65% for vitamin A and E, respectively. And, for overweight/obese women the inadequacy was identified in 44% and 31% for these micronutrients, and 44% for vitamin B2 (Table 4)”. I am certain that you can improve the grammar of these paragraph but my point is to focus on the inadequacy rates.
Response:
We have replaced
“More than 50% of women met the H-AR for vitamin A, thiamine, riboflavin, vitamin B6, vitamin B12, vitamin C, vitamin E (except for lean women), calcium, magnesium (except for overweight/obese women), iodine and iron through diet alone. More than 87% of women did not meet the H-AR for niacin, folate and vitamin D through diet alone (Table 4).
Dietary supplements increased the proportion of pregnant women meeting the H-AR (Table 4), notably for folate, vitamin D, magnesium, and vitamin E which reached greater than 50% H-AR (except for vitamin E in the lean group) when supplements were taken.”
With
“Deficiencies were notable for niacin, folate and vitamin D as more than 77%, 95% and 95% of women, respectively, did not satisfy the H-AR dietary requirements with the consumption of foods only. The number of deficiencies were reduced for folate and vitamin D with supplements, but the deficiencies remained high in a significant proportion of women. Deficiencies in the other micronutrients were not as severe but still notable for vitamin A (35% in lean, 44% in overweight/obese); riboflavin (44% overweight/obese); vitamin E (77% lean, 50% overweight/obese); iodine (46% lean, 37% overweight/obese); iron (44% overweight/obese) and magnesium (50% lean, 56% overweight/obese). The number of deficiencies in these nutrients were improved with supplements, however, a large proportion of women still did not meet the H-AR for vitamin E (65% lean, 31% overweight/obese) and magnesium (31% overweight/obese).
Dietary supplements increased the proportion of pregnant women meeting the H-AR (Table 4), notably for folate, vitamin D, iodine, iron, magnesium, and vitamin E which reached greater than 50% H-AR (except for vitamin E in the lean group) when supplements were taken.”
Lines 302 and 303, and lines 316-319, of the discussion should also be modified to match with the description of the dietary inadequacies as mentioned above.
Response:
Lines 302 and 303 - We have replaced “More than 87% of all women did not meet the H-AR for niacin, folate and vitamin D through diet alone”
With
“More than 87% of all women did not meet the H-AR for niacin; more than 95% of women did not meet the H-AR for folate and vitamin D through diet alone.”
Lines 316-319 – we have replaced “Micronutrients, such as niacin, folate and vitamin D were consumed in very low amounts and >87% of women did not meet the H-AR for these micronutrients from diet alone.”
With
“Micronutrients, such as niacin, folate and vitamin D were consumed in very low amounts and more than 87%, 95% and 95% of women did not meet the H-AR for niacin, folate and vitamin D, respectively from diet alone.”
Finally, similarly to the comment that vitamin D can be synthesized through exposure of the skin to sunlight, you can add a sentence saying that niacin may be synthesized using the amino-acid tryptophan as a precursor.
Response:
We have added the suggestion regarding being able to synthesise niacin to the paragraph that discusses Folate and Niacin and NTD. Therefore we added the following “Unlike folate, niacin is not an essential nutrient as humans can synthesise niacin from the amino acid tryptophan, where it is estimated that 60mg of dietary tryptophan is equivalent to 1mg niacin [35].”
Due to the addition of one reference, we have amended the reference list and the associated numbering of references in the manuscript.
Also, to differentiate between folate being an essential nutrient that we must obtain from our diet, and niacin and vitamin D can be synthesised by humans, we have added the fact that folate is an essential nutrient in the introduction in the second paragraph. Therefore we have added “, which is an essential nutrient and must be obtained through the diet as humans cannot synthesise folate.” To the first sentence of the second paragraph which now reads “There are certain nutrients of relevance to pregnancy that must be obtained from the diet and a notable example is folate, which is an essential nutrient and must be obtained through the diet as humans cannot synthesise folate.”
Reviewer 2 Report
Comments and Suggestions for Authors
All the questions were answered.
Author Response
Reviewer 2
All the questions were answered.
Response: We thank the reviewer for their comment.